# Cuticular modified air sacs underlie white coloration in the olive fruit fly, *Bactrocera oleae*

Manuela Rebora [1✉], Gianandrea Salerno [2✉], Silvana Piersanti[1], Alexander Kovalev [3] & Stanislav Gorb [3✉]

Here, the ultrastructure and development of the white patches on thorax and head of *Bactrocera oleae* are analysed using scanning electron microscopy, transmission electron microscopy, and fluorescence microscopy. Based on these analyses and measurements of patch reflectance spectra, we infer that white patches are due to modified air sacs under transparent cuticle. These air sacs show internal arborisations with beads in an empty space, constituting a three-dimensional photonic solid responsible for light scattering. The white patches also show UV-induced blue autofluorescence due to the air sac resilin content. To the best of our knowledge, this research describes a specialized function for air sacs and the first observation of structural color produced by tracheal structures located under transparent cuticles in insects. Sexual dimorphism in the spectral emission also lays a structural basis for further investigations on the biological role of white patches in *B. oleae*.

[1] Dipartimento di Chimica, Biologia e Biotecnologie, University of Perugia, Via Elce di Sotto 8, Perugia, Italy. [2] Dipartimento di Scienze Agrarie, Alimentari e Ambientali, University of Perugia, Borgo XX Giugno, Perugia, Italy. [3] Department of Functional Morphology and Biomechanics, Zoological Institute, Kiel University, Am Botanischen Garten 9, Kiel, Germany. ✉email: manuela.rebora@unipg.it; gianandrea.salerno@unipg.it; sgorb@zoologie.uni-kiel.de

Insects exhibit a remarkable diversity of colours which represent important visual cues for intra and inter-specific communication being used in sexual recognition or as disruptive or warning coloration against visually oriented predators[1,2]. Such a diversity of colours can derive from pigments, either embedded in the cuticle or situated under a transparent cuticle[3], or can be produced through structural optical systems from selective light reflection[4].

Structural colours are widespread in insects[5,6] and can arise from interference, diffraction and/or scattering of incident light owing to the presence of nanostructures characterizing insect cuticular surface and its multi-layered structure. Structural colours, sometimes associated with pigments[7,8], have been extensively studied, especially in some insect orders such as Lepidoptera[9,10], Coleoptera[11] and Odonata[7,12–14] which offer remarkable examples in this field.

Structural whiteness in insects is less common than colours and requires scattering processes for all visible wavelengths[15]. White light may be scattered by unpigmented cuticle, setae, scales and surface waxes with specific shapes and size, thus producing structural white. This has been documented in beetles, due to a three-dimensional photonic solid in the scales of *Cyphochilus* spp.[15]; in butterflies (Pieridae), due to an array of microscopic beads suspended within the wing scales[16,17] and in Odonata, owing to the presence of wax crystals on the epicuticle[18–21]. In particular, beetle's white scales optimized to produce optical scatter and highly efficient whiteness, brilliance, and opacity with an extremely thin thickness have been extensively studied[22–24].

In the present study, we analyze, in detail, the mechanism underlining the production of structural white in the olive fruit fly *Bactrocera oleae* (Rossi) (Diptera: Tephritidae). Among Tephritidae, which are of major economic importance in agriculture, this fly is a key pest in olive crops worldwide and represents a great problem, especially in the Mediterranean basin[25,26]. Adults of both sexes bear, on their thorax and head, white patches (yellowish in dead or old specimens), whose biological function is still unknown. Ultrastructural investigations using scanning (SEM) and transmission electron microscopy (TEM), together with observation in light and fluorescence microscope as well as reflection spectra measurements allowed us to shed light on the nature of the structures responsible for light scattering. Among different white patches, located on the thorax and head, we focused our observations mainly on the scutellum, a small triangular plate placed posteriorly in the mesonotum, because it is wider and more easily, both microscopically and experimentally, accessible than the other white areas.

## Results

The thorax of the adult *B. oleae* shows the following white patches: the scutellum, two post pronotal lobes, two anepisternal areas, two notopleural calli, and two katatergites (Fig. 1). In addition to these, white areas are also located at the front of the head (Fig. 1, inset).

### Development and ultrastructure of the white patches in *Bactrocera oleae* adults.

Observations under a stereomicroscope allowed us to discover that the white patches of the *B. oleae* thorax and head are visible under the transparent cuticle. To describe the development of white areas, we mainly focused on the scutellum patch and observed that the pharate adult does not show any white coloration (Fig. 2a), while the just emerged adult shows a scutellum with a transparent cuticle, under which two separate reduced white areas are visible (Fig. 2b). In the hours following emergence, these white areas tend to increase their size (Fig. 2c) until their development under the transparent cuticle is

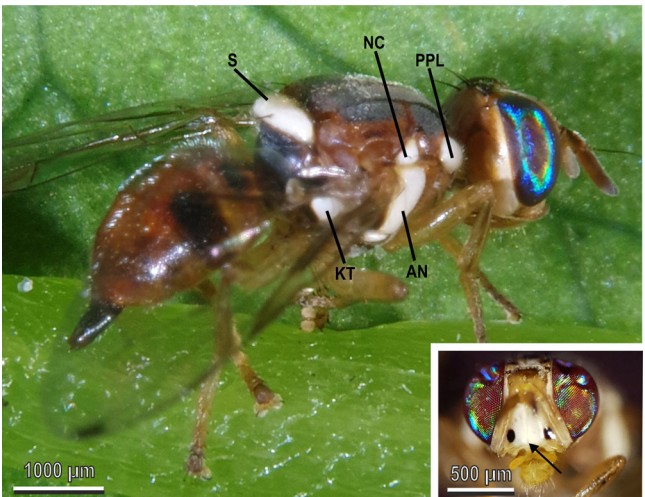

**Fig. 1 Adult (female) of *Bactrocera oleae* with white patches.** Thoracic white patches of the olive fruit fly: S scutellum, PPL post pronotal lobe, AN anepisternal area, NC notopleural callus, KT katatergite. Inset shows white patches on the head (arrow).

completed, which is realised 24 h after emergence (Fig. 2d), thus giving the entire scutellum a bright white colour.

Semithin sections of the scutellum (Fig. 3) reveal its internal structure composed of two air sacs that, in the pharate adult, are still compressed and reduced in size (Fig. 3a). In the just emerged adult, the two air sacs begin to inflate and their lumen increases even if an empty space is still present under the transparent cuticle of the dorsal side of the scutellum (Fig. 3b). In the adult (starting from 24 h after emergence), the two air sacs are fully developed under the transparent cuticle (Fig. 3c–f) and white. In the fully developed air sacs, the transparent cuticle and the epidermis are in close contact with a layer of vesicles (developed during the pharate stage) under which the air sac is present (Fig. 3c–f). In this dorsal area, the internal side of the air sac is modified and shows arborisations (Fig. 3c–f), already visible in the pharate adult (Fig. 3a) and in the just emerged adult (Fig. 3b). The arborisations of the air sac and the vesicular layer are located just under the transparent cuticle on the dorsal side of the scutellum while they disappear in the dark-pigmented area (Fig. 3c–f). Male and female show sexual dimorphism in the development of the vesicular layer: females (Fig. 3c, e) tend to have a thicker vesicular layer than that of males (Fig. 3d, f). The thickness of the vesicular layer of the female is $7.5 \pm 1.9\,\mu m$ ($n = 6$) (mean ± SD), which is significantly different ($t = 4.90$, d.f. = 9, $p = 0.0008$) from that of the males measuring $2.9 \pm 0.8\,\mu m$ ($n = 5$) (mean ± SD). There is no significant difference ($t = 0.38$, d.f. = 8, $p = 0.712$) between the two sexes in the maximal thickness of the arborisations of the air sac which is $15.4 \pm 2.4\,\mu m$ (mean ± SD) in females ($n = 6$) and $16 \pm 2.9\,\mu m$ (mean ± SD) in males ($n = 5$).

Cryo SEM revealed that the scutellum of ten-day-old adults of *B. oleae*, is similar to other white patches present in the fly, and is constituted of a smooth cuticle with reduced or absent microtrichia in comparison with other body areas (Fig. 4a, b). The transparent cuticle located in these white patches is made up of different layers (Fig. 4c) under which air sacs are present. The air sacs just under the transparent cuticle show prominent arborisations (Fig. 4d–h). Such arborisations run perpendicularly to the cuticle and measure about 15 μm in length (Fig. 4e, f). Each arborisation is characterised by the presence of numerous spiny beads with a diameter of about 0.35 μm (Fig. 4g). The thick layer of arborisations in the air sacs is situated only under the white patches (Fig. 4h), while under the dark cuticle no arborisation is

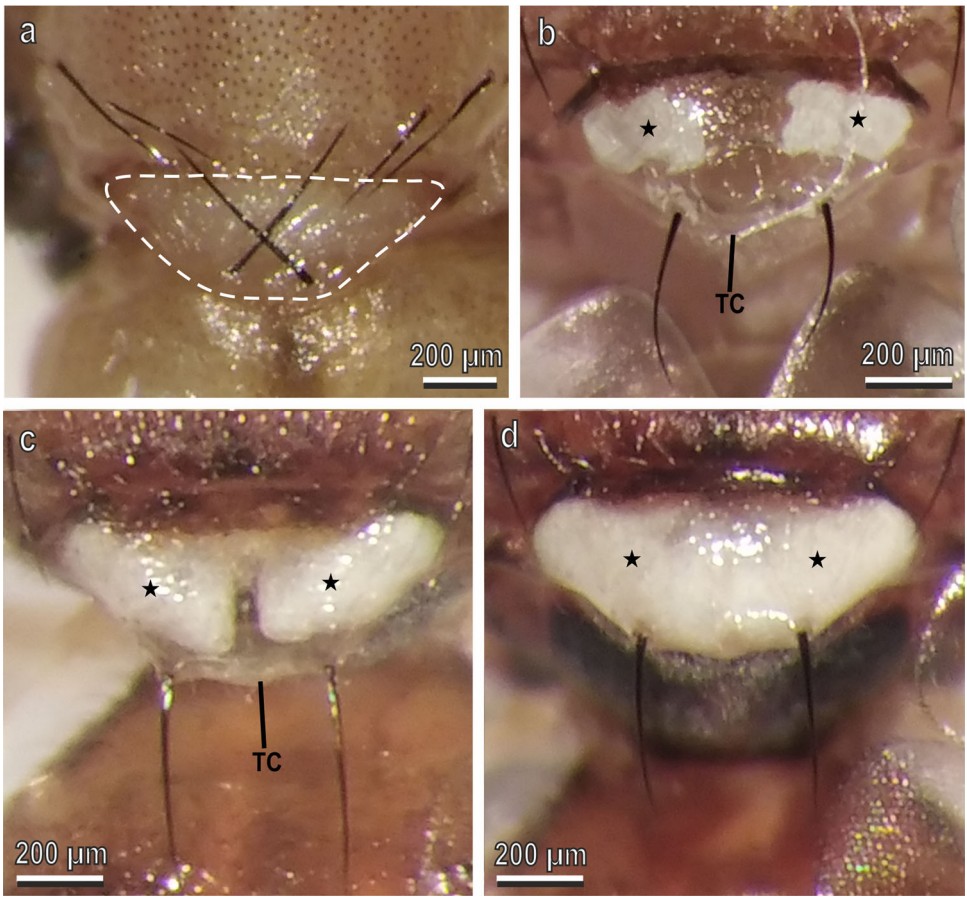

**Fig. 2 Dorsal view of the scutellum of *Bactrocera oleae* under stereomicroscope. a** Scutellum (dotted line) in the pharate adult without any white coloration; **b**, **c** Scutellum in the just emerged adult (**b**) and in the adult six hours after emergence (**c**) with transparent cuticle (TC) under which two separated reduced white areas are visible (stars), corresponding to the two air sacs; **d** Scutellum in the adult 24 h after emergence with transparent cuticle, under which two fully developed air sacs (stars) are present, giving it a uniform bright white colour.

present and the walls of the air sac show only rows of thick cuticle with spiny beads (Fig. 4d, h).

Sections of the scutellum visualised with TEM allowed describing in detail the development and the structure of the arborisations of the air sacs in correspondence to the white patches (Fig. 5 and Supplementary Figs. 1–3). The pharate adult shows a scutellum with the cuticle and epidermis underlined by a vesicular layer, under which haemolymph and the air sac are visible (Supplementary Fig. 1a). The air sac is constituted of a monolayer of large cells with a basal membrane and an apical portion, from which arborisations depart towards the lumen of the air sac (Supplementary Fig. 1a–d). These cells are characterised by large nuclei, rough endoplasmic reticulum and mitochondria (Supplementary Fig. 1b). Their apical border shows plaques at the tip of microvilli secreting the cuticular layer lining the arborisations (Supplementary Fig. 1d). Such arborisations are constituted of cytoplasm bordered by a thin epicuticular layer and numerous spiny beds (Supplementary Fig. 1c). The epicuticular layer borders the beads, located at the apex of stems constituted of procuticle (Supplementary Fig. 1d). In the pharate adult, the spiny beds are developing: they are not yet uniformly electron-dense but appear rich in granules (Supplementary Fig. 1d).

The scutellum of the just emerged adult shows a multi-layered transparent cuticle with the underlying epidermis, a developed vesicular layer, under which haemolymph and the air sac are visible (Supplementary Fig. 2a). The monolayer of big cells, constituting the air sac in correspondence to the dorsal side of the scutellum under the transparent cuticle, is well developed

(Supplementary Fig. 2b) and shows a thick layer of arborisations lined by cuticle (Supplementary Fig. 2c). At this stage, the spiny beads are fully developed and appear very electron-dense (Supplementary Fig. 2d). They are located on stems constituted of procuticle lining cells of the air sac together with the epicuticular layer (Supplementary Fig. 2d).

In the ten-day-old adult (Fig. 5a), the main difference in comparison with the just emerged adult, is the thickness of the monolayer of big cells constituting the air sac. Indeed, as reported above, in the hours following emergence, the air sacs inflate, reaching their final size by means of the distension of the air sac cells. The cells constituting the air sac (Fig. 5b–c) show well-developed nuclei and cytoplasm organelles (mitochondria, Golgi apparatus), but they are very flattened in comparison with the pharate adult and the just emerged adult. The amount of cytoplasm in the cells of the air sac is reduced, especially in the arborisations (Fig. 5b–d) which show mainly vesicles within them (Fig. 5d). The electron-dense spiny beds are numerous and randomly distributed and oriented along all the arborisations (Fig. 5b). Their cuticular stems appear thinner and elongated in comparison with the pharate adult and the just emerged adult (Fig. 5d). The length of the arborisations are reduced, moving far from the dorsal side of the air sac until their disappearance. Only scattered spiny beads are visible along the inner side of the air sac bordering the lumen (Fig. 5e). The vesicular layer under the epidermis, is visible just over the developed arborisations of the air sacs, located under the white patches of *B. oleae* and absent under the dark-pigmented cuticle (Figs. 5e, 3c–f). Such a vesicular

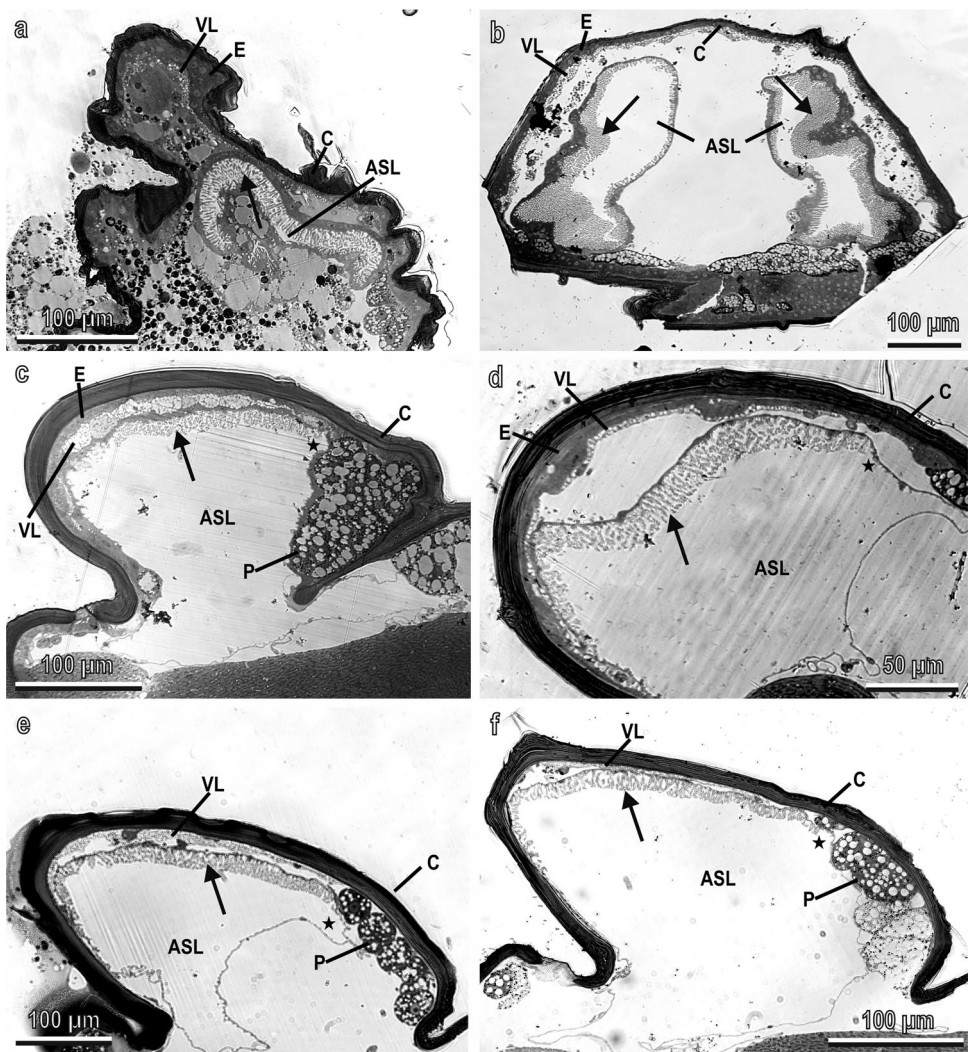

**Fig. 3 Semithin sections of the scutellum of *Bactrocera oleae*. a** Scutellum in the pharate adult. Longitudinal section showing an air sac still compressed and reduced in its lumen (ASL); **b** Scutellum in the just emerged adult. Frontal section showing the two air sacs with their lumen (ASL) which begin to inflate increasing in size. Note the central empty space still present under the transparent cuticle (C); **c–f** Scutellum in the female three days after emergence (**c**) and ten days after emergence (**e**) and in the male three days after emergence (**d**) and ten days after emergence (**f**). Longitudinal sections showing one of the two air sacs fully developed under the transparent cuticle (C). Note that the arborisations of the air sac and the vesicular layer are located just under the transparent cuticle on the dorsal side of the scutellum while they disappear in the pigmented area (P). Asterisk points out the transition between the arborisations with the vesicular layer and the pigmented area; Arrow points out the arborisations; ASL air sac lumen, C cuticle, E epidermis, P pigmented area, VL vesicular layer.

layer is constituted of cells with cytoplasm full of electron-translucent vacuoles, already visible in the pharate adult (Supplementary Fig. 3a, b). The vesicle amount tends to increase from the pharate adult to the just emerged adult (Supplementary Fig. 3a–c). In the adult, the cell cytoplasm has almost disappeared and a uniform thick layer of electron-translucent vesicles is visible (Supplementary Fig. 3d, e). As reported above, the vesicular layer in the female (Supplementary Fig. 3d) tends to be thicker than in the male (Supplementary Fig. 3e).

**Fluorescence properties**. The scutellum and the other white patches on both the thorax and head of *B. oleae*, show UV-induced fluorescence (Fig. 6). The excitation occurs at 365 nm (UV light) and the emission from 397 nm (blue light). Such fluorescence is very visible 24 h after emergence, when the air sacs are fully developed. It is weaker in the just emerged adults, where the air sacs are still reduced in size and not inflated (see Supplementary Fig. 4). No clear difference has been highlighted in the

fluorescence emitted by males and females (see Supplementary Fig. 4). A weaker fluorescence (green or red) has been observed with other filters (excitation at 450–490 nm and emission from 520 nm and excitation at 546 nm and emission from 590 nm (see Supplementary Fig. 5).

**Reflectance spectra**. The reflectance spectra of white patches in *B. oleae* thorax are relatively flat in the range of 480–1000 nm wavelengths. However, there is a broad and low amplitude maximum at around 830 nm. Reflection in UV range is considerably lower (4.8 and 6.2 times lower for 45° and 0°/30° observation direction, respectively), decreases to the lower range of wavelengths, and demonstrates multiple low amplitude maxima within 220–320 nm (Fig. 7, Supplementary Table 1, Supplementary data 1). The weak scattering wavelength dependence and low scattering in UV range confirm that the scattering takes place on particle sizes larger than the wavelength.

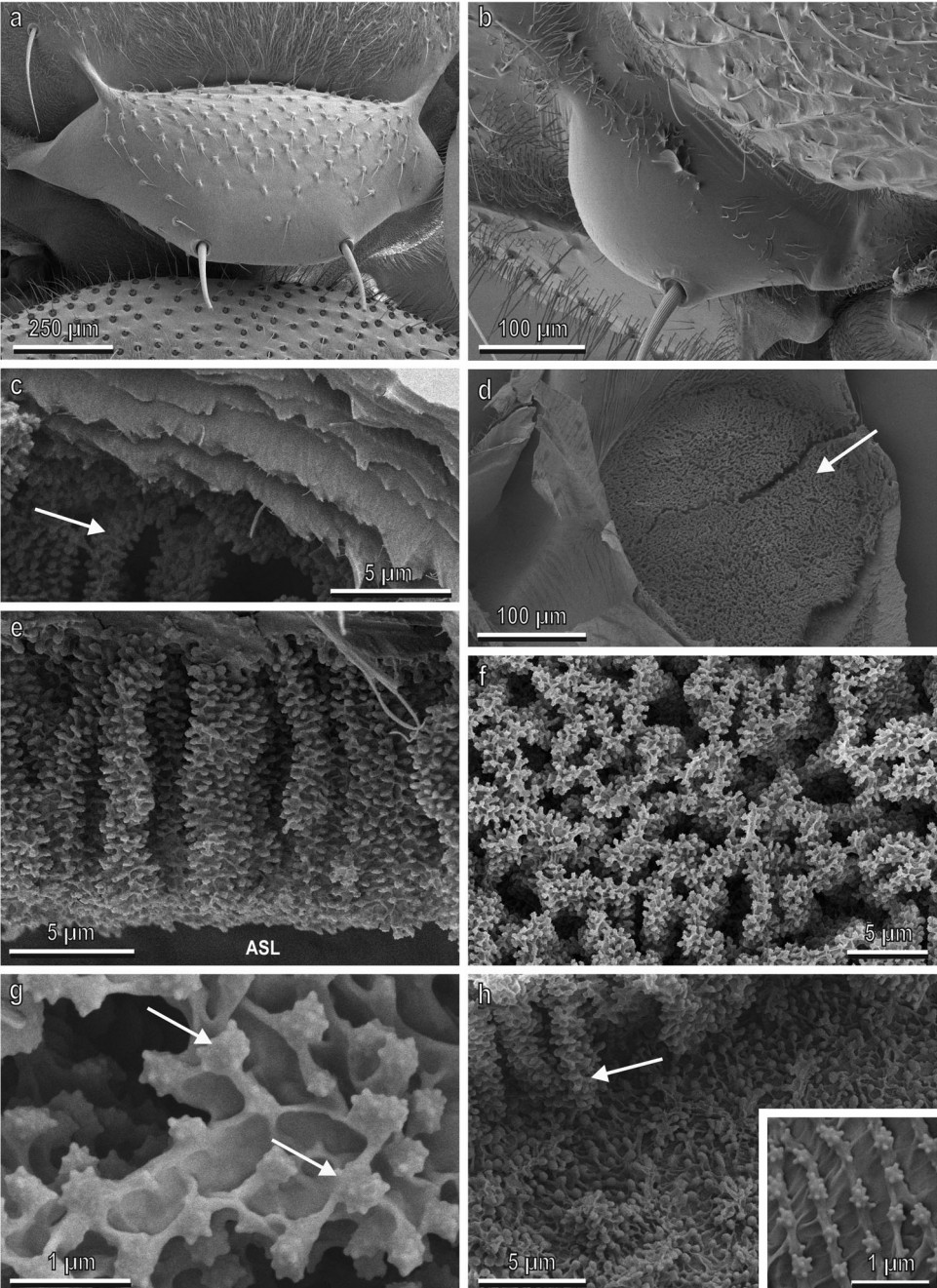

**Fig. 4 Scutellum of adults of *Bactrocera oleae* ten days old visualised with Cryo SEM. a** Dorsal vision of the scutellum; **b** Dorsal view of the notopleural callus. Note the cuticular surface with reduced or absent microtrichia in comparison with other body areas; **c** Detail of the multi-layered transparent cuticle located on the white thoracic patches under which air sacs with arborisations (arrow) are present; **d** Internal vision of the air sac located under the katatergite. Note the prominent arborisations under the transparent cuticle (arrow); **e** Lateral view of the long arborisations of the inner portion of the air sac facing the air sac lumen (ASL) under the transparent cuticle; **f** Detail of the arborisations visualised from the internal side (lumen) of the air sac; **g** Detail of the arborisations characterised by the presence of numerous spiny beads (arrows); **h** Detail of (**d**) showing that the arborisations of the air sacs (arrow) are present only under the transparent cuticle (corresponding to the white patches) while, in correspondence of the dark cuticle, no arborisation is present and the walls of the air sac show only scattered beads.

Reflection from the dark areas on the *B. oleae* thorax was used as a reference for the interpretation of the reflection properties of the white patches. The dark areas demonstrated almost Lambertian reflection through the whole measured spectral range. However, since the cuticle in those areas is melanised and relatively smooth, the reflection intensity in dark areas on the thorax is three times lower than that in white spot area (in visible spectral range) (Fig. 7, Supplementary Table 1). The cuticle in white patches lacks melanin, and therefore it is transparent. Yet the cuticle roughness of the thorax is more or less homogeneous, the specular reflection in UV spectral range in white patches is lower than in the dark areas (0.56 of the reflection in the dark areas) (Fig. 7, Supplementary Table 1, Supplementary Table 2).

The reflection from white patches is close to the angle-independent Lambertian reflection. Thus, after observation angle correction the reflection intensities from white spots are 0.246,

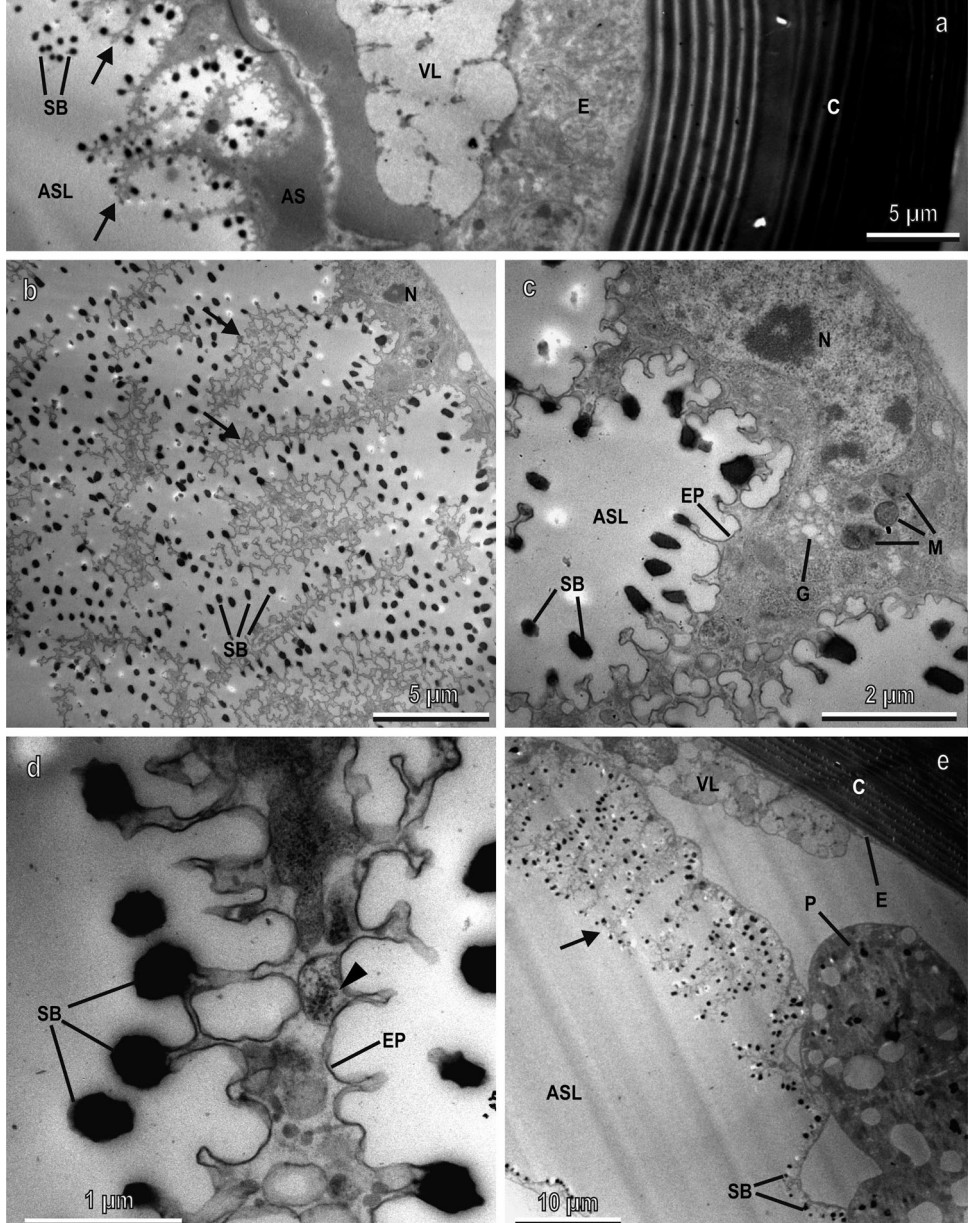

**Fig. 5 Longitudinal sections of the scutellum of ten days old adult (female) of *Bactrocera oleae* visualised with TEM. a** General view showing the multi-layered transparent cuticle (C), the epidermis (E), the vesicular layer (VL), and the air sac (AS) constituted of a monolayer of thin cells with arborisations (arrows) bordered by electron-dense spiny beads (SB). ASL air sac lumen; **b**, **c** Details of one of the cells of the air sac with its arborisations (arrows) and spiny beads (SB). BL basal lamina, EP epicuticular layer, G Golgi apparatus, M mitochondria, N nucleus; **d** Detail of one of the arborisations containing granules and vesicles (arrow heads) with very electron-dense spiny beads (SB). Note that the cuticular stems of the beads (arrow) appear thin and elongated. EP epicuticular layer; **e** View of the scutellum moving far from the dorsal area, showing that the length of arborisations is reduced and only scattered spiny beds (SB) are visible facing the air sac lumen (ASL). Note that the vesicular layer (VL) is visible just over the developed arborisations of the air sacs (arrow). (P). C cuticle, E epidermis.

0.267, 0.262 (in males) and 0.218, 0.228, 0.219 (in females) for observation angles 0°, 30°, and 45°, respectively (Fig. 7, Supplementary Table 1, Supplementary Table 3). The white spots in males reflect ~16% more visible light than in females. Besides, the variation in reflection by individual females is slightly (1.5 times) higher than by individual males.

In the experiments to measure the reflected and transmitted light of the thoracic white patches of *B. oleae* in different conditions: intact, with the air sac removed, intact infiltrated with oil and with the air sac removed infiltrated with oil (Fig. 8, Supplementary data 2–3) we could observe that intact white patches reflect light, but absorb transmitted light (Fig. 8a, b, i, j); when the subcuticular air sac is carefully removed, the cuticle becomes transparent in both reflected and transmitted lights showing a high light transmittance and a low light absorption (Fig. 8c, d, i, j); when intact white patches are infiltrated with oil with refractive index more similar to the cuticle material as air, after 30 min the cuticle becomes transparent in both reflected and transmitted lights showing a diminished light scattering in comparison with the same condition without oil (Fig. 8e, f, i, j); when the subcuticular air sac is removed and cuticle embedded in oil with refractive index more similar to the cuticle material as air,

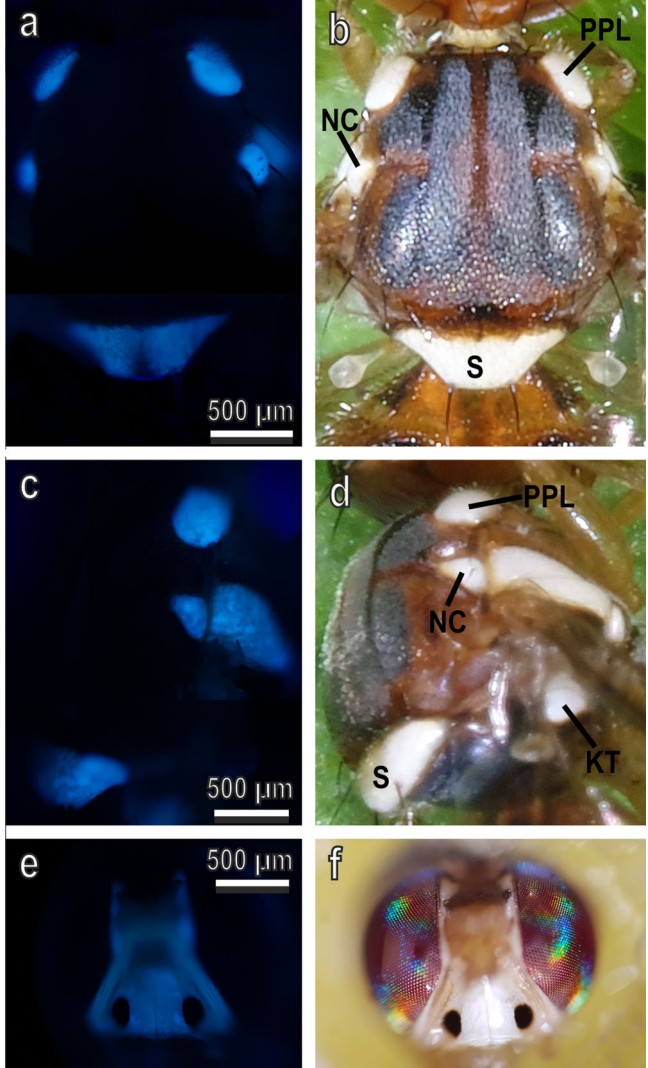

**Fig. 6 Thorax and head of *Bactrocera oleae* in fluorescence light microscope and bright-field light microscope.** Dorsal (**a**, **b**) and lateral (**c**, **d**) view of the thorax and frontal view of the head (**e**, **f**) of an adult (ten days old) in fluorescence light microscope with an excitation filter 365 nm, chromatic beam splitter FT 395 nm, emission 397 nm (**a**, **c**, **e**) and bright-field light microscope (**b**, **d**, **f**). Note the UV-induced fluorescence of the white patches on the thorax. S scutellum, PPL post pronotal lobe, AN anepisternal area, NC notopleural callus, KT katatergite.

after 30 min, the cuticle becomes transparent in both reflected and transmitted lights (Fig. 8g–j), with almost no difference in the light transmittance and absorption in comparison with the same condition without oil.

## Discussion

The present investigation describes in detail the ultrastructure of the white patches visible in the thorax and in the head of *B. oleae*. Our data reveal that only in correspondence of these white patches (1) the cuticle is transparent lacking melanisation; (2) under the epidermis, the fly air sacs assume a peculiar structure with developed arborisations; (3) a highly vacuolated layer is interposed between the cuticle and the air sacs.

The peculiar arborisations of the air sacs under the white patches, with numerous cuticular spiny beads randomly distributed and oriented in an empty space with air, have features highly compatible with the production of structural white. Indeed

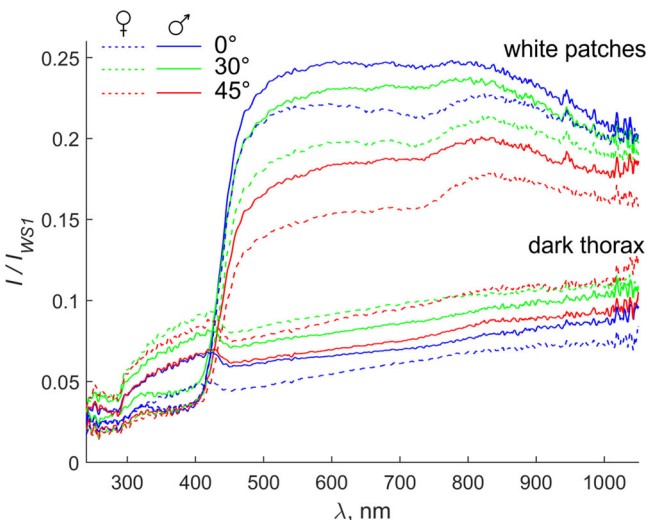

**Fig. 7 Mean reflectance spectra from white patches and dark areas on thorax of *Bactrocera oleae*.** Illumination was at 45°, and observation was at 0° (blue), 30° (green), and 45° (red) to the surface normal. Reflection intensity is presented as a ratio to the reflection from WS1 reflectance standard. The reflection spectra from the dark thorax area of females/males are the lower lines series (in the range 500–1000 nm), the reflection spectra from the white patches on thorax of females/males are shown by upper lines series. Spectra from males are shown as a solid line, female spectra are shown as a dashed line.

the perception of whiteness is due to scattering from a material containing many disordered surfaces reflecting all wavelengths[5]. Structural whiteness in bird feathers or in insects is usually produced by empty structures made of a continuous biopolymer phase and air: voided spaces contain air, while the other phase is a solid material, such as chitin or beta-keratin[4]. Among insects, the wing whiteness of the cabbage white butterflies (Pieridae) is produced by microscopic beads suspended within the wing scale[16,17], while the body whiteness of the scarab beetles of the genus *Cyphochilus* is produced by a random network of interconnecting cuticular filaments inside thin scales showing high scattering efficiency[15,22,27]. Accordingly, the arborisations of the aerial sacs of *B. oleae* with their epicuticular layer and the cuticular spiny beads randomly oriented could constitute a three-dimensional photonic solid surrounded by air able to produce structural white. Indeed, strong diffuse reflection in white spot areas, which is ~20% of the reflection of fine WS1 standard, might be related to the modified air sacs described here. The involvement of these structures in producing structural white is highly suggested by our experiments measuring the reflected and transmitted light of the thoracic white patches of *B. oleae* in different conditions: intact, with the air sac removed, intact infiltrated with oil and with the air sac removed infiltrated with oil (Fig. 8). The specular reflection in the visible spectral range measured at 45° from the patch with the air sac removed was 0.8 of that in intact white patch. The scattering takes place on particles of about 0.35 μm in size similar to the diffuse reflection coming from submicrometer-sized chitin filaments network in scales of *Cyphochilus* spp. and *Lepidiota stigma* F. (Coleoptera: Scarabaeidae) beetles[23].

The immersion oil used to infiltrate the air sacs has a high refractive index, which is similar to the refractive index of insect cuticle, $n \approx 1.56$[28,29] (we used the refractive index of the beetle cuticle even if the actual refractive index of the white patches of the olive fruit fly is unknown). If the light scattering takes place on the interface between the cuticle and air, it should be

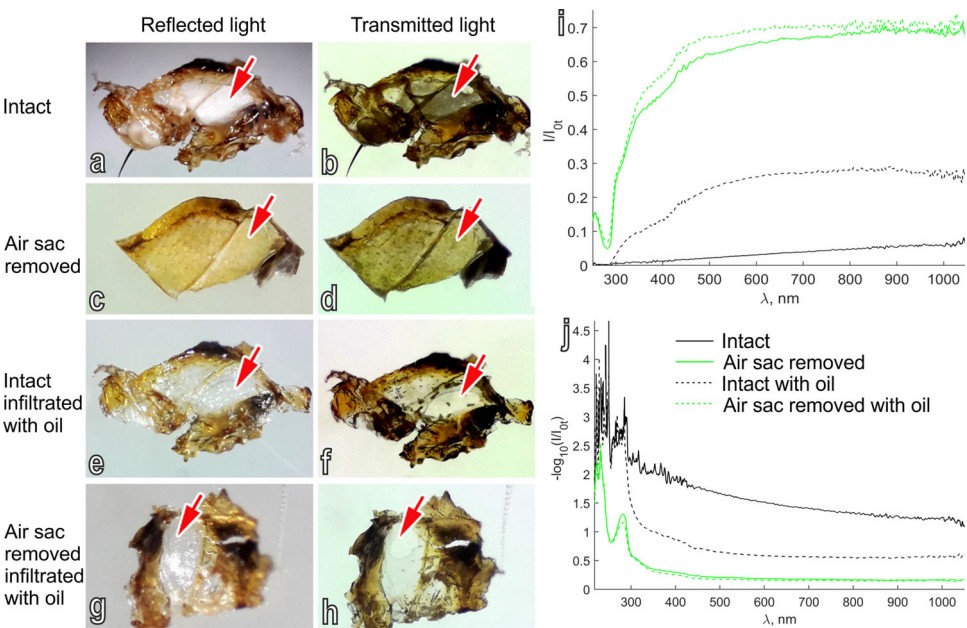

**Fig. 8 Thoracic white patches of *Bactrocera oleae* photographed with reflected and transmitted light.** Thoracic white patches photographed with epi- (reflected light) (**a**, **c**, **e**, **g**) and trans- (transmitted light) (**b**, **d**, **f**, **h**) illumination, in different conditions: (1) intact (**a**, **b**); (2) with the underlining air sac removed (**c**, **d**); (3) intact, infiltrated with oil (**e**, **f**); and (4) with the underlining air sac removed, infiltrated with oil (**g**, **h**). Transmission (**i**) and absorption (**j**) spectra of the white patches in the different conditions, intact (black solid line); with the underlining air sac removed (green solid line); intact, infiltrated with oil (black dashed line); and (4) with the underlining air sac removed, infiltrated with oil (green dashed line).

suppressed, when the oil fills the interconnected air voids in the modified air sacs. Light transmittance from the white patches samples infiltrated with oil is, however, lower than the transmittance of the cuticle with air sac removed. Remaining air voids or intracuticular defects/inclusions could be responsible for this reduced transmittance.

The reflection from white patches in males, which is significantly higher than in females, could be due to the different development (statistically different) of the electron-translucent vesicular layer in the two sexes observed over the air sac arborisations, but further investigations are necessary to clarify this point.

The white patches in the body of males and females of *B. oleae*, in particular the scutellum, the two post pronotal lobes, the two anepisternal areas, the two notopleural calli, the two katatergites together with the white area in the head, are located just in correspondence of the air sacs, expansions of the tracheae, which in Diptera are located mainly in the thorax and head[30] (see scheme in Fig. 9). Typically, insect air sacs inflate during 24 h after the emergence[31]. In agreement with this, as observed in our experiments, the structural white in the scutellum of *B. oleae* is uniformly present under the transparent cuticle only 24 h after the emergence. In the pharate adult, no white coloration is visible, in agreement with the not yet fully inflated air sac. In the pharate adult, the spiny beds and the arborisations of the air sac are still not fully developed, thus contributing to the absence of white patches in this stage.

The typical functions of aerial sacs in insects are to increase tracheal respiratory efficiency, to assist flight by lowering the specific gravity, hydrostatic function in aquatic insects, thermoregulation, amplification and resonance for sound production/ reception[31]. The present investigation highlighted a further function for aerial sacs in the production of structural colour, thus highlighting once again the astonishing evolutionary adaptive plasticity of insects able to modify pre-existence structures for new purposes. This feature could be present also in other arthropods since in the scutigeromorph species *Scutigera*

*coleoptrata* L. (Scutigeromorpha: Scutigeridae) tracheal bundles appear as white spots under transparent cuticle[32]. The reflected white is not as pronounced as in *B. oleae*, but it could represent a possible preliminary stage for further specialization in insects. In *B. oleae* the spiny beads located along the arborisations of the air sacs under the white patches could derive from the papillae (punctuate thickenings) described along the inner surface of insect trachee[33,34] and of fly air sacs[35]. The tracheal cuticle in insects consists of a thin epicuticle and a conspicuous procuticle and follows regular protrusions of the apical plasma membrane of tracheal epithelial cells from which taenidia derive[36]. Most probably, taenidia, papillae and spiny beads have same evolutionary origin and the same chemical composition of procuticle and epicuticle, also suggested by our ultrastructural observations. In this regard, it is interesting to remember that a peculiar kind of procuticle with an additional protein with the characters of keratins was suggested in insect tracheal taenidia[37].

The white patches in the body of *B. oleae* emit fluorescence as highlighted in our observations under fluorescence microscope. In particular, the stronger excitation occurs at 365 nm (UV light) and the emission from 397 nm (blue light).

In nature, autofluorescence has been reported in numerous taxa (see reviews in refs. [38,39]), including flowers, plants, corals, worms, squid, tardigrades, spiders, stomatopods, insects, fish, reptiles, birds, and can be used as a means of intersexual communication[40], prey capture[41], camouflage[42], or as a protection against UV radiation[43]. Among insects, some species belonging to the groups of beetles, ants, butterflies, dragonflies and one bee species revealed to emit fluorescence (review in ref. [44]). In insects, fluorescence is often due to the presence of pteridines, metabolic products of purines absorbing light around 360 nm and fluorescing in the blue region[45]. As in *B. oleae*, in many cases, insect fluorescence is not dispersed uniformly on the body surface, but it is localised in some areas, usually the same areas that have markings colored with natural pigments contrasting with the rest of the body, such as the beetles, *Pachyrhynchus gemmatus purpuerus* Kraatz (Coleoptera: Curculionidae)

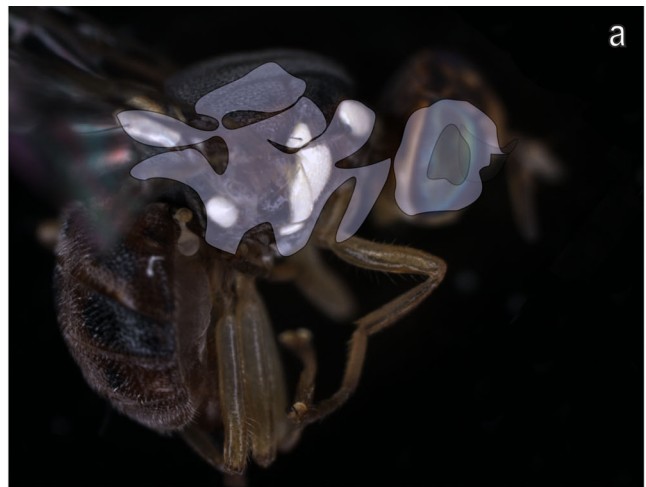

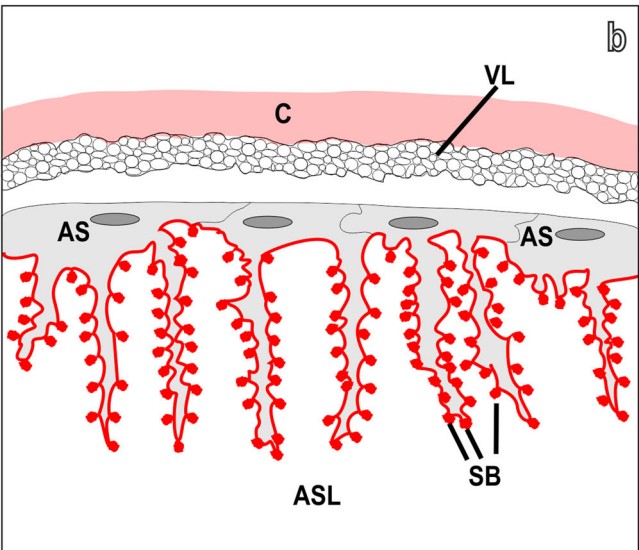

**Fig. 9 *Bactrocera oleae* air sacs and schematic structure of their modified internal structure. a** Development of air sacs (in grey, modified from ref. [33]) in relation with the distribution of white patches; **b** Scheme showing the structure of the modified air sac (AS) with arborisations and spiny beads (SB) underlining the transparent cuticle (C) in correspondence of the white patches. VL, vesicular layer. The red line shows the thin layer of epicuticle bordering the arborisations and the spiny beads facing the air sac lumen (ASL) filled with air.

or *Cicindela maritima*[44] Dejean (Coleoptera: Carabidae). Indeed, the characteristics of the insect surface on which the fluorophore is present can significantly affect the intensity and distribution of the emitted radiation and of the visual signal in the observer[38,46]. In this regard, it is interesting to remember the case of the beetles *Celosterna pollinosa sulfurea* Buquet (Coleoptera: Cerambycidae) and *Phosphorus virescens* Oliver (Coleoptera: Cerambycidae), where fluorescence occur in conjunction with three-dimensional photonic structures[47]. Similarly, the butterfly *Papilio nireus* L (Lepidopotera: Papilionidae) shows in the scales of its wings a two-dimensionally periodic photonic crystal infused with a highly fluorescent pigment represented by pterin. Such photonic crystals enhance the intensity of the observed fluorescence from the butterfly's wing[46].

An explanation of the origin of autofluorescence under white patches of *B. oleae* might be the presence of resilin (an autofluorescent protein with an emission in the blue region) in the cuticle of the air sac. The strong argument for this is that the UV-excited autofluorescence in the blue region of the dissected

thoracic air sacs is similar to that emitted by the white patches of *B. oleae* (while in other wavelengths is not present or rather low (see Supplementary Fig. 6)). In tracheae (and consequently in air sacs), the presence of di- and trityrosines, which serve as indicators for resilin, has been previously demonstrated[48]. Resilin is presumably responsible for elasticity/resilience of tracheal tubes and air sacs. The trityrosine-to-dityrosine ratio in the various cuticular regions vary from nearly equal amounts of the two amino acids to about ten times more dityrosine than trityrosine, indicating that the regions differ in degree of cross-linking, but the tracheal wall is the material with the highest trityrosine-to-dityrosine ratio[48]. From the biomechanical point of view, it would mean that this tracheal resilin must be the stiffest type of this rubber-like protein. The detected UV-excited blue fluorescence through the cuticular windows of the patches in *B. oleae* might be a resilin-driven side effect. This is in agreement with the reflection/scattering of the white patches in UV range which is low (Fig. 7) and lower in the white areas than in the dark areas (0.56 of the reflection in the dark areas). The measured absorption spectra demonstrate that the white patches contain proteins, since characteristic peaks for aromatic amino acids (223 nm; tryptophan, 280 nm; tyrosine, 274 nm)[49] and peptide bond (230 nm)[50] can be clearly seen (Fig. 8). Such a UV light absorption in the white patches of *B. oleae* is in agreement with the presence of amino acids constituting resilin, a protein highly represented in insect air sacs.

Interestingly electrophysiological testing of the spectral sensitivity of *B. oleae* adult visual system revealed a major sensitivity peak at 485–500 nm (blue light)[51], thus supporting the hypothesis of a possible biological role of the white patches in *B. oleae* life.

To our knowledge, no attention has been previously paid to the possible biological role of the white patches on the body of *B. oleae*. Similar to the observations in *Drosophila*[52], olive fruit flies that emerged in winter show a darker body colour, compared with flies that emerged during summer, but this concerns only the melanised portions of the body while the white patches on the thorax and head remain unchanged in summer and winter[53].

Concerning the sexual differences observed in the reflectance spectra of the scutellum (the reflection from white patches in males is higher than those in females), one possible explanation could be a role of white patches as visual cues in intra and intersexual recognition during mating. In this regard, it is important to remember that investigations on the recognition of potential rivals/partners during mating behaviour in *B. oleae* (and in Tephritidae in general) are focused mainly on chemical cues, while to the best of our knowledge, no studies have elucidated the behavioural role of visual cues in lek defence, courtship and mating behaviour and further research on this point is required[54].

It has been previously demonstrated that two predatory fly species, *Lispe consanguinea* Loew (Diptera: Muscidae), and *L. tentaculata* DeGeer (Diptera: Muscidae), inhabiting the supra-littoral zone at the shore of a fresh-water reservoir possess similar visual cues namely reflective concave silvery scales on the head face[55]. Flies occupy different lek habitats. Males of the first species patrol the bare wet sand on the beach just above the surf. Males of the second species reside on the more textured heaps of algae and stones. Male dancing is usually evenly distributed around the female. Rival males also circle about one another at a distance shorter than 15 mm, but not in close contact. Flies of both species often walk sideward and observe the partner not in front but at the side[55]. We hypothesize that the frontal and lateral patches in *B. oleae* might have similar function in the courtship of this species and in representatives of other Tephritidae in general.

Another possible function of the white patches could be linked to predator avoidance. Indeed, morphological and behavioural aposematic mimicry with jumping spiders has been reported in

other Tephritidae species belonging to the genus *Rhagoletis*[56] and *Zonosemata*[57,58]. Interestingly Salticidae females have palps with a UV-excited bright fluorescence[40] similar to that observed in the white patches of *B. oleae*. Imitation of jumping spiders could be of benefit against spiders, insects and vertebrate fly predators since jumping spiders are difficult to catch and poisonous.

Finally, the white patches of *B. oleae* could potentially serve other non-visual functions, such as thermoregulation. In beetles of the species *Neocicindela perhispida* (Broun) (Coleoptera: Cicindelidae), white morphs were capable of foraging for a longer time without overheating, when transferred from their natural white sand habitat to a black sand terrain, compared with the resident black morphs, which escaped the heat burrowing into the sand[59].

The data here presented describe in detail the ultrastructure and optical properties of the white patches on the body of the olive fruit fly *B. oleae*. The patches have important implications. First, structural white and associated autofluorescence are described in Diptera. Second, a structural colour not produced by an external cuticle but by an internal structure located under transparent cuticle is described in Insects. In particular, portions of the air sacs with their modified internal structure assume a function that, to the best of our knowledge, is so far undescribed in insects. Third, the identification of these complex structures producing structural white lays the foundation for further investigations aiming to understand the biological role of the white patches on the body of *B. oleae* and their possible use as visual cues in sex recognition or predatory avoidance. This knowledge, adding information on the biology of this dangerous species, could potentially help to develop methods for its biological control.

## Methods

**Insects**. *B. oleae* adults emerged from pupae obtained from olives collected in the field around Perugia (Umbria, Italy) during October in 2019 and 2020. Olives were kept in the laboratory in a controlled condition chamber (14 h photophase, temperature of $25 \pm 1\,^\circ\text{C}$; RH of $60 \pm 10\%$), on a net, in order to collect fallen pupae. Pupae were kept inside net cages ($300\,\text{mm} \times 300\,\text{mm} \times 300\,\text{mm}$) under the same controlled conditions until adult emergence. Both females and males, after emerging, were maintained in the cages and provided with water and crystallised sucrose.

**Light microscopy**. To describe the development of the white patches, we performed observations under a stereomicroscope. Observations of the scutellum of adults at different ages—(1) adults, which have completed the metamorphosis but were still within the pupa (pharate adults), (2) just emerged adults, (3) adults, six hours after emergence, (4) adults, 24 h after emergence—were performed with a stereomicroscope Leica MZ6 (Leica Microsystem GmbH, Wetzlar, Germany).

Semithin sections of the scutellum *of B. oleae* were obtained from the thorax of anaesthetized insects at different ages (pharate adult, just emerged adult, three days old adult, ten days old adult). Specimens were fixed for 3 h in 2.5% glutaraldehyde in cacodylate buffer (Electron Microscopy Sciences, Hatfield, England), pH 7.2. The fixed scutelli were repeatedly rinsed in sodium cacodylate buffer and post-fixed for 1 h at 4 °C in 1% osmium tetroxide in sodium cacodylate buffer (Electron Microscopy Sciences, Hatfield, England). The samples were then repeatedly washed in the same buffer, dehydrated in ascending ethanol concentrations and finally embedded in an Epon-Araldite resin mixture (Sigma-Aldrich). Afterwards, semi-thin sections (1 μm thick) were sectioned using a Leica EM UC6 ultra microtome (Leica Microsystem GmbH, Wetzlar, Germany), stained with methylene blue and observed in a Leica DMLB light microscope (Leica Microsystem GmbH, Wetzlar, Germany).

To calculate the mean thickness of the vesicular layer (six measurements for each sample) and the maximal length of the arborisations in the two sexes, measurements were made from digital images with the open-source image processing program ImageJ[60].

**Fluorescence microscopy**. Observations of the thoracic white patches of adults at different ages—(1) just emerged adults, (2) three-day-old adults, (3) ten-day-old adults—were performed using a fluorescence microscope Zeiss Axiophot (Zeiss, Jena, Germany) with an excitation filter 365 nm, chromatic beam splitter FT 395 nm, emission 397 nm, with an excitation filter 450–490 nm, chromatic beam splitter FT 510 nm, emission 520 nm, and with an excitation filter 546 nm,

chromatic beam splitter FT 580 nm, emission 590 nm. Live males and females were anaesthetised with carbon dioxide, fixed with tape to a microscope slide and observed in the fluorescence microscope. To analyse the autofluorescence of dissected air sacs located inside the thorax (usually covered by dark cuticle) and compare it to that produced by the intact scutellum (with transparent cuticle), the thorax of a fly has been dissected under a stereomicroscope, placed on a glass slide and observed using a fluorescent microscope with the three excitation filters mentioned above.

**Cryo scanning electron microscopy (Cryo-SEM)**. The shock-frozen samples of the thorax of *B. oleae* insects (ten-day-old males and females) were studied in a scanning electron microscope (SEM) Hitachi S-4800 (Hitachi High-Technologies Corp., Tokyo, Japan) equipped with a Gatan ALTO 2500 cryo-preparation system (Gatan Inc., Abingdon, UK). For details of sample preparation and mounting for cryo-SEM, see ref. [61]. Pieces of the thorax with white patches were carefully cut out, mechanically clamped to the metal holder and shock-frozen in liquid nitrogen. After transferring the frozen sample to the cryo-chamber of the microscope, it was fractured with a metal blade, cooled at −140 °C, and sputter-coated under frozen conditions with gold-palladium (thickness 10 nm) and examined at 3 kV acceleration voltage and a temperature of −120 °C on the cryo-stage within the microscope. In addition, we prepared fractures of the corresponding sites in dry insects and viewed them using conventional SEM.

**Transmission electron microscopy (TEM)**. The scutellum of 2 pharate adults, 4 just emerged adults (2 females and 2 males), 4 three day old adults (2 females and 2 males) and 8 ten-day-old adults (4 females and 4 males) of *B. oleae* were dissected from anaesthetized insects and fixed for 3 h in 2.5% glutaraldehyde in cacodylate buffer (Electron Microscopy Sciences, Hatfield, England), pH 7.2. The fixed scutelli were repeatedly rinsed in sodium cacodylate buffer and post-fixed for 1 h at 4 °C in 1% osmium tetroxide in sodium cacodylate buffer (Electron Microscopy Sciences). The samples were then repeatedly washed in the same buffer, dehydrated in ascending ethanol concentrations and finally embedded in an Epon-Araldite resin mixture (Sigma-Aldrich). Afterwards, ultra-thin sections were cut using a Leica EM UC6 ultramicrotome (Leica Microsystem GmbH, Wetzlar, Germany), collected on Parlodion (Spi-Chem, West Chester, PA, USA) coated copper grids, stained with uranyl acetate and lead citrate (Electron Microscopy Sciences, Hatfield, England) and examined using a Technai G2 Spirit BioTwin Transmission Electron Microscope (FEI company, Hillsboro, OR, USA) equipped with a CCD camera Veleta (EMSIS) $2048 \times 2040$.

**Reflectance spectra measurements**. The dorsal side of the white patches (scutellum) of adult *B. oleae* (ten days old) were illuminated at 45° using a light source (DH-2000-BAL, Ocean Optics Inc, Dunedin, Florida, USA) through an optical fibre (200 μm diameter) equipped with a one lens condenser. The condenser was placed 10 mm away from the sample. The reflected light was collected by another condenser, which was placed 35 mm away from the sample, and could collect the light at various observation angles (0°, 30°, and 45° for our measurements), and it was mounted on an optical fibre (200 μm diameter). The optical fibre was connected to a spectrometer (Ocean Optics Inc, Dunedin, Florida, USA). The spectra were recorded with the software Spectral Suite (Ocean Optics Inc, Dunedin, Florida, USA). Further spectra processing was performed in Matlab 7.10 (The MathWorks, Natick, NA, USA) and includes the dark noise subtraction, smoothing with Savitzky-Golay filter[62] of 4th order with 12 nm window, and normalization on the WS1 standard (Ocean Insight, Ostfildern, Germany) reflection spectrum (45° illumination/detection) (Supplementary Fig. 7, Supplementary data 4). No observation angle correction was performed. The mean intensity in the UV range was calculated within 330–400 nm, the mean intensity in red spectral range was calculated within 550–700 nm. Using this arrangement, the reflectance spectra measurements of the scutellum and dark areas on the thorax of 5 females and 5 males were done.

We observed the thoracic white patches of *B. oleae* with epi- (reflected light) and trans- (transmitted light) illumination, in different conditions: (1) intact, (2) with the underlining air sac removed, (3) intact, infiltrated with oil and (4) with the underlining air sac removed and infiltrated with oil. In particular, a thoracic white patch was cut from the insect thorax using a razor blade and photographed with epi- and trans-illumination in different conditions. Transmission and reflection spectra were also recorded in different conditions. The air sac, underlining the white patch, was scratched out using a piece of a razor blade. A tiny amount of oil (Immersion oil TM 518, Carl Zeiss, Oberkochen, Germany; $n = 1.518$) was applied to the cuticle with and without subcuticular air sacs.

**Statistics and reproducibility**. The thickness of the vesicular layer and the maximal thickness of the arborisations of the air sac in males and females were compared using a Student's $t$ test for independent samples. Three-way ANOVA analysis on reflection measurements was carried out using SigmaPlot 12.5 (Systat Software GmbH, Erkrath, Germany). For post hoc analysis the Holm-Sidak method was used.

**Reporting summary**. Further information on research design is available in the Nature Research Reporting Summary linked to this article.

## Data availability

Source data underlining graphs (Figs. 7, 8i, j, Supplementary Fig. 7) are included in this published article (and Supplementary Data 1–4). All other data are available from the corresponding author on reasonable request.

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

## Acknowledgements

The research was performed thanks to the knowledge acquired during training stages at the Department of Functional Morphology and Biomechanics of the University of Kiel supported by Erasmus grants (Staff mobility for training 2019-20 (M.R.) and (G.S.)).We are very grateful to "Laboratorio di Microscopia Elettronica, Dipartimento di Scienze della Vita, Università degli Studi di Siena" and in particular to Prof. Romano Dallai and Eugenio Paccagnini, for the use of the Transmission Electron Microscope. Victoria Kastner from Max Planck Institute of Developmental Biology, Tübingen, Germany is greatly acknowledged for English corrections of the manuscript.

## Author contributions

The study was designed by all the authors. S.G. performed the cryo-scanning electron microscopy investigations. M.R. and S.P. performed the transmission electron microscopy investigations. G.S. performed the light microscopy and the fluorescence microscopy investigations. A.K. performed the reflectance spectra observations. The manuscript was written by M.R., G.S., A.K. and S.G. All authors discussed the analysis and interpretation of the results and participated in the final editing of the manuscript.

## Funding

## Competing interests

The authors declare no competing interests.
