## [Peer Review File · Communications Biology]

Reviewers' comments:

Reviewer #1 (Remarks to the Author):

The manuscript of Reborá et al. describes their study of the white patches observed in male and female fruit flies *Bactrocera oleae* which feed on olive fruits. The authors report on their experimental analysis of the structural whiteness based on light microscopy, fluorescence microscopy, cryo-SEM, TEM, and reflectance spectra measurements. The overall analysis, however, is solid but more descriptive. As summarized by the authors in their introduction there are several other insects with famous white structure like the white beetle of the genus *Cyphochilus*. In these studies the structures causing whiteness were analyzed in great detail and compared to other structures causing broadband scattering. This aspect is missing completely in the actual study. This makes the actual study less meaningful. Therefore, I think the manuscript should be published in a more specialized journal after considering the following issues:

- The analysis of the reflectance in Fig. 10 is poor. The data is presented in arbitrary units and not comparable to other studies. Moreover, I am wondering that the reflectance for wavelengths below 420 nm is larger for the dark areas than for the white patches. Furthermore, even if we consider that the data is presented in arb. units it is surprising to me that the difference in reflectance is so low between white and dark areas. In the methods section the authors mention that they smooth the data using a Savitzky-Golay filter. I think such an approach is highly questionable for the presented case because such a smoothing flattens the data, i.e., pushed it towards desired shape of a flat curve. Finally, I highly recommend using some kind of standard reference to quantify the whiteness. The described approach with MgCO₃ powder in the UV makes a comparison with other studies quite difficult.
- Some sentences in the introduction are easy to be misunderstood, i.e.,
Line 55: "... with periodic features ..." is a limitation. In many cases structural colors are produced by non-periodic or even random structures. Please rephrase.
Line 61: "White light may be scattered by unpigmented cuticle, setae, scales and surface waxes, thus producing structural white." -> unpigmented cuticle, setae, surface waxes do not produce whiteness without proper shape and size, thus I think the last part of the sentence "thus producing structural white" should be rephrased.
- The potential of the examined structure of biomimetic application is stressed out in the introduction and in the conclusion. However, no specific idea is given how this might happen. Please, give concrete examples/ideas or omit this statement.
- In Lines 209 to 217 the authors present some nanometer values down to the second position after decimal point and reference Figure 3. I wonder how this accuracy might be obtained from the images presented in that figure? Please explain.
- Throughout the manuscript, the "air sacs" and "spiny beads" are mentioned and described. However, the reader does not learn so much about the following: How do these structures produce whiteness? How is their size distribution and orientation with respect to the insect body? Here, a sketch might be helpful. Finally, what is the material of the "spiny beads". The manuscript states that they are "electron dense" but that's a very limited information. To my opinion the proper description of the structures producing the observed whiteness is a key issue to make the whole study meaningful.
- A smaller issue: I recommend avoiding statements like the one in Line 445 "... for the first time ...". If it's true, the readers will notice by themselves.

Reviewer #2 (Remarks to the Author):

The authors described the white patches in *Bactrocera oleae* analyzed using an SEM and a TEM and fluorescence microscope, and measured its reflection spectrums of white area and the dark thorax. The major claim of the paper is the white reflection was caused by the structure inside of the white patches. The point of them to reveal the mechanism of which patches might be quite interesting for the others in the wider field. However, their results were only morphological data and reflection measurements. It is hard to believe the white reflection is caused by the inner structure of the white patches. It seems the white reflection is caused by the relation between the

air and the spiny beads, but no evidence of it. And also there is no research on the mechanism of structural color.

From a scientific point of view, I think that the authors should add at least the following experiments.

1. They should collect a few insects from the field and measure the spectral back reflection of the white patch part in vivo. Surgically remove only the white patch cuticle and measure the spectral back reflection and the transmission spectrum. If the cuticle is transparent in this experiment, they can speculate that the internal structure of the white patch is involved in the creation of color.

2. They should excise the cuticle of the white patch surgically with such as a razor and measure the reflection spectrum from the inside of the white patch. With this experiment, they can strongly speculate that the white color is caused by the inner structure.

*Another method; If they have the transverse section of white patch without staining, they can measure the spectral absorption curves both of the cuticle and inside structure using a micro spectral photometer.

By the experiments of 1 and 2 described above, the authors can finally reach their conclusion that it is a structural color. I know that it is not easy to collect the olive fruit fly in winter and also due to the influence of COVID-19, but I think that it should not be said that it is a structural color without the above experiment. If it is difficult to collect, they must prove that the structure is colored white by using another species of insect with a similar white patch structure or by engineering imitating the white patch structure.

As long as the title clearly states "structural white", it is necessary to prove that the white reflection is a structural color by the other scientific methods.

Major;

1. The reflection curve of white patches (Fig.10) showed the broad curve above 400 nm. Why the reflectance between ca. 250 to ca. 400 nm of white patches was so low that the dark thorax. Do the white patches possess a certain absorption material such as pigments which absorb mainly shorter wavelength regions?

2. To describe the origin of structural color, the authors only used the morphological data. It is hard to understand the mechanism only from the morphology in the white patches. The authors should show more clearly how to determine the white color was caused by the structure.

Minor;

1. The authors measured from the different angles. To show the whiteness, it is good to use their technique, but why do they not show the reflection curve obtained by back reflection. There is a possibility to show the contribution of the multilayered structure of the surface of the cuticle.

2. Page 6 L140; They should show a brief description how to prepare the crack (cut) sample to observe the inside of the white patches.

3. Why did they used MgCO₃ for the reference instead of the white reference?

4. In Fig.6, how did the authors distinguish between HE and ASL in this morphological data?

5. If possible, please measure the absorbance and transparent spectrum of the cuticle of the white patches. Please note that the strongest reflection is usually came from the surface of the cuticle.

Reviewer #1 (Remarks to the Author):

The overall analysis, however, is solid but more descriptive.

We would like to thank the reviewer for the valuable suggestions. Analysis was extended.

-As summarized by the authors in their introduction there are several other insects with famous white structure like the white beetle of the genus *Cyphochilus*. In these studies the structures causing whiteness were analyzed in great detail and compared to other structures causing broadband scattering. This aspect is missing completely in the actual study. This makes the actual study less meaningful.

The structured color in *B. oleae* appears because of the undercuticular structures (the modified air sacs). The cuticle itself is transparent. Manuscript title is changed to emphasize the novelty of the study.

-The analysis of the reflectance in Fig. 10 is poor. The data is presented in arbitrary units and not comparable to other studies.

We replaced Fig. 10 in order to make it more comparable to other studies (using WS1 standard for normalization of reflection spectra).

-Moreover, I am wondering that the reflectance for wavelengths below 420 nm is larger for the dark areas than for the white patches.

White patches of *Bactrocera oleae* might absorb the UV light owing to the presence of resilin in the air sacs. We better clarified this aspect in the Results and in the Discussion section.

-Furthermore, even if we consider that the data is presented in arb. units it is surprising to me that the difference in reflectance is so low between white and dark areas.

We agree, indeed the dark areas could be called grey areas. We added a sentence in the Result section: "... since the cuticle in those areas is melanised and relatively smooth, the reflection intensity in dark areas on thorax is three times lower than that in white spot area (in visible spectral range)."

-In the methods section the authors mention that they smooth the data using a Savitzky-Golay filter. I think such an approach is highly questionable for the presented case because such a smoothing flattens the data, i.e., pushed it towards desired shape of a flat curve.

The spectra obtained with and without application of Savitzky-Golay filter are shown in Fig. 1supplementary.

-Finally, I highly recommend using some kind of standard reference to quantify the whiteness. The described approach with MgCO₃ powder in the UV makes a comparison with other studies quite difficult.

Reflection spectrum of WS1 standard was used for normalization of reflection spectra. It is formulated in Methods section: “Further spectra processing was performed in Matlab 7.10 (The MathWorks, Natick, NA, USA) and includes the dark noise subtraction, smoothing with Savitzky-Golay filter (Fig. 1supplementary), and normalization on the WS1 standard (Ocean Insight, Ostfildern, Germany) reflection spectrum (45° illumination/detection).” Fig. 10 is replaced.

- Some sentences in the introduction are easy to be misunderstood, i.e., Line 55: “... with periodic features ...” is a limitation. In many cases structural colors are produced by non-periodic or even random structures. Please rephrase.

We removed “with periodic features”.

-Line 61: “White light may be scattered by unpigmented cuticle, setae, scales and surface waxes, thus producing structural white.” -> unpigmented cuticle, setae, surface waxes do not produce whiteness without proper shape and size, thus I think the last part of the sentence “thus producing structural white” should be rephrased.

We rephrased the sentence as follows: “White light may be scattered by unpigmented cuticle, setae, scales and surface waxes with specific shape and size“

- The potential of the examined structure of biomimetic application is stressed out in the introduction and in the conclusion. However, no specific idea is given how this might happen. Please, give concrete examples/ideas or omit this statement.

We added a sentence at the end of the Discussion: “Structural white colors are widely used in many industries, especially in those of paints, plastics, and ceramics. For such applications, thermal stability, chemical stability, dispersibility, color strength, light fastness, and opacifying power are required. This is the main reason, why instead of using classical pigments, structural colour pigments are increasingly taken into consideration in the selection of a suitable pigment (as an example, see patent application US20040156986A160). We believe that geometrical parameters of the structures studied here, their arrangement in natural system(s), and protection of the delicate structure responsible for the color formation could be used to develop improved white colored materials with tailored properties.”

- In Lines 209 to 217 the authors present some nanometer values down to the second position after decimal point and reference Figure 3. I wonder how this accuracy might be obtained from the images presented in that figure? Please explain.

We are sorry for the mistake, they are micron instead of nanometers (it was a typo mistake). We corrected it. In any case we rounded the values.

- Throughout the manuscript, the “air sacs” and “spiny beads” are mentioned and described. However, the reader does not learn so much about the following: How do these structures produce whiteness? How is their size, distribution, and orientation with respect to the insect body? Here, a sketch might be helpful.

We introduced a new figure (Fig. 12) to show the development of air sacs inside the thorax of *B. oleae* in relation with the distribution of white patches. Moreover, in the same figure, we reported a scheme with the structure of the modified air sac underlining the transparent cuticle in correspondence of the white patches.

-Finally, what is the material of the “spiny beads”. The manuscript states that they are “electron dense” but that’s a very limited information. To my opinion the proper description of the structures producing the observed whiteness is a key issue to make the whole study meaningful.

We added the following sentence in the Discussion “Most probably, taenidia, papillae and spiny beads have same evolutionary origin and the same chemical composition of procuticle and epicuticle, also suggested by our ultrastructural observations. In this regard it is interesting to remember that a peculiar kind of procuticle with an additional protein with the characters of keratins was suggested in insect tracheal taenidia (Baccetti et al., 1984).

- A smaller issue: I recommend avoiding statements like the one in Line 445 “... for the first time ...”. If it’s true, the readers will notice by themselves.

If the referee agrees, we would like to keep this statement because we think it helps to highlight some important aspects, such as the novelty of an air sac used in the production of structural color. We would like to stress the functional plasticity of insect organs and not all the readers of a multidisciplinary journal as *Biology Communications* know the typical function of insect air sacs.

Reviewer #2 (Remarks to the Author):

From a scientific point of view, I think that the authors should add at least the following experiments.

1. They should collect a few insects from the field and measure the spectral back reflection of the white patch part in vivo. Surgically remove only the white patch cuticle and measure the spectral back reflection and the transmission spectrum. If the cuticle is transparent in this experiment, they can speculate that the internal structure of the white patch is involved in the creation of color.

2. They should excise the cuticle of the white patch surgically with such as a razor and measure the reflection spectrum from the inside of the white patch. With this experiment, they can strongly speculate that the white color is caused by the inner structure.

*Another method; If they have the transverse section of white patch without staining, they can measure the spectral absorption curves both of the cuticle and inside structure using a micro spectral photometer.

By the experiments of 1 and 2 described above, the authors can finally reach their conclusion that it is a structural color. I know that it is not easy to collect the olive fruit fly in winter and also due to the influence of COVID-19, but I think that it should not be said that it is a structural color without the above experiment. If it is difficult to collect, they must prove that the structure is colored white by using another species of insect with a similar white patch structure or by engineering imitating the white patch structure.

As long as the title clearly states “structural white”, it is necessary to prove that the white reflection is a structural color by the other scientific methods.

We would like to thank the reviewer for the valuable suggestions. Unfortunately, our setup does not allow measuring the back reflection. Instead we performed the following experiments and added a new Figure (Fig. 11):

We observed the thoracic white patches of *B. oleae* with epi- (reflected light) and trans- (transmitted light) illumination, in different conditions: 1) intact, 2) with the underlining air sac removed, 3) intact, infiltrated with oil and 4) with the underlining air sac removed, infiltrated with oil. In particular, a thoracic white patch was cut from the insect thorax using razor blade and photographed with epi- and trans-illumination in the different conditions. Transmission and reflection spectra were also recorded in the different conditions. The air sac, underlining the white patch, was scratched out using a piece of a razor blade. A tiny amount of oil (Immersion oil™ 518, Carl Zeiss, Oberkochen, Germany; n=1.518) was applied to the cuticle with and without subcuticular air sacs.

Conclusions are (see Result section): intact white patches reflect light, but absorb transmitted light (Fig. 11a,b,i,j); when the subcuticular air sac is carefully removed, the cuticle is transparent in both reflected and transmitted lights showing a high light transmittance and a low light absorption (Fig. 11c,d,i,j); when intact white patches are infiltrated with oil with refractive index more similar to the cuticle material as air, after 30 min the cuticle is transparent in both reflected and transmitted lights, showing a diminished light scattering, in comparison with the same condition without oil (fig. 11e,f,i,j); when the subcuticular air sac is removed and cuticle embedded in oil with refractive index more similar to the cuticle material as air, after 30 min, the cuticle is transparent in both reflected and transmitted lights (fig. 11g,h, i,j), with almost no difference in the light transmittance and absorption in comparison with the same condition without oil.

1. The reflection curve of white patches (Fig.10) showed the broad curve above 400 nm. Why the reflectance between ca. 250 to ca. 400 nm of white patches was so low that the

dark thorax. Do the white patches possess a certain absorption material such as pigments which absorb mainly shorter wavelength regions?

"The reflection/scattering in UV-range is low. Thus, the light absorption should be responsible for transmission decay in this range and absorption could be calculated from transmittance. The absorption spectra demonstrate the white spot area contains proteins, since characteristic peaks for aromatic amino acids (223 nm; tryptophan, 280 nm; tyrosine, 274 nm) [Takahashi et al., 1981] and peptide bond (230 nm) [Saraiva, 2020] can be clearly seen there. The presence of these proteins is in agreement with the presence of resilin, typically occurring in air sacs. White patches of *Bactrocera oleae* might absorb the UV light owing to the presence of resilin in the air sacs located under the transparent cuticle. We better clarified this aspect in the Results and in the Discussion section.

2. To describe the origin of structural color, the authors only used the morphological data. It is hard to understand the mechanism only from the morphology in the white patches. The authors should show more clearly how to determine the white color was caused by the structure.

Strong diffuse reflection in white spot areas, which is ~20% of the reflection of fine WS1 standard, might be related to the subcutical modified air sacs. It was demonstrated in additional experiments (see referee 2) with removal of air sacs under the transparent cuticle of white patches and their oil infiltration (see Fig. 11). The involvement of these structure in producing structural white is highly suggested by our experiments measuring the reflected and transmitted light of the thoracic white patches of *B. oleae* in different conditions: intact, with the air sac removed, intact infiltrated with oil and with the air sac removed infiltrated with oil (Fig. 11). The specular reflection in visible spectral range measured at 45° from the patch with the air sac removed was 0.8 of that in intact white patch. The scattering takes place on particles of about 0.35 μm in size similarly to the diffuse reflection coming from submicrometer sized chitin filaments network in scales of *Cyphochilus* and *Lepidiodia stigma* beetles²⁵.

Minor;

1. The authors measured from the different angles. To show the whiteness, it is good to use their technique, but why do they not show the reflection curve obtained by back reflection. There is a possibility to show the contribution of the multilayered structure of the surface of the cuticle.

Unfortunately, our setup doesn't allow to measure the back reflection. However, the measurements with illumination at -45° and detection at 45° were done.

2. Page 6 L140; They should show a brief description how to prepare the crack (cut) sample to observe the inside of the white patches.

Done. See M&M section.

3. Why did they used MgCO₃ for the reference instead of the white reference?

Reflection spectrum of WS1 standard was used for normalization of reflection spectra. It is formulated in Methods section: “Further spectra processing was performed in Matlab 7.10 (The MathWorks, Natick, NA, USA) and includes the dark noise subtraction, smoothing with Savitzky-Golay filter (Fig. 1s), and normalization on the WS1 standard (Ocean Insight, Ostfildern, Germany) reflection spectrum (45° illumination/detection).” Fig. 10 is replaced.

4. In Fig.6, how did the authors distinguish between HE and ASL in this morphological data?

HE refers to haemolymph (it is outside the air sac and contains scattered blood cells) while ASL is the air sac lumen. It is very easy to distinguish between these two compartments. In insects, as known, inside the air sacs (air sac lumen) there is no haemolymph but air. For the production of structural white it is important to have empty structures made of a continuous biopolymer phase and air such as that realised in the air sacs of *Bactrocera oleae*.

5. If possible, please measure the absorbance and transparent spectrum of the cuticle of the white patches. Please note that the strongest reflection is usually came from the surface of the cuticle.

Transmission and reflection spectra were also recorded”.

It is true for specular reflection of ideally flat surface. “The specular reflection in the visible spectral range measured at 45° from the cuticle with the air sac removed was 0.8 of that in native state.”.

Reviewers' comments:

Reviewer #1 (Remarks to the Author):

The revised manuscript of Reborá et al. describes their study of the white patches observed in male and female fruit flies *Bactrocera oleae* which feed on olive fruits. In the revision the authors consider some of the issues mentioned by the two referees. They mainly considered optical spectroscopy issues (a white standard is used, and the smoothing procedure is explained in more detail). However, several issues were not considered, and I still think that the manuscript would be better suited for a more specialized journal.

- There are still some issues in the introduction. In my review, I gave one example which is considered by the authors ("with periodic features"). However, when I wrote that some sentences in the introduction are easy to be misunderstood, I assumed that the authors would take this polite hint to re-edit the complete text. So, I will be more direct now: Please note that the whole manuscript contains several very general statements which are not scientifically true. An example, line 61 should read "... and requires scattering of all VISIBLE wavelength". One might consider that tiny details, but please do your readers a favor and re-edit the manuscript carefully.

- The discussion is extended with some more details about potential applications. However, I think the arguments are weak. The industry seems to be very happy with conventional TiO₂ to achieve whiteness. The added patent does not fit to whiteness. It deals with color pigments and is abandoned. You might want to consider that TiO₂ is considered a potential harm for health and the environment recently. That's the reason why many industries look for replacements and other researchers focused on structural whiteness. I have no problem if there is no potential application. The results might be still of great scientific interest. However, if you mention bio-inspiration in the abstract there should be a serious discussion about that in the main text.

- In my review I wrote: "A smaller issue: I recommend avoiding statements like the one in Line 445 "... for the first time ...". If it's true, the readers will notice by themselves.

Your answer is: " If the referee agrees, we would like to keep this statement because we think it helps to highlight some important aspects, such as the novelty of an air sac used in the production of structural color. We would like to stress the functional plasticity of insect organs and not all the readers of a multidisciplinary journal as *Biology Communications* know the typical function of insect air sacs."

I disagree. The term "For the first time" is mentioned in the abstract and on page 20 in the discussion. The phrase "new" is mentioned with the same meaning two times in the abstract, additionally on page 16, and two times on page 20. I assume the readers will get the message if such claims are less frequently found in the manuscript.

Reviewer #2 (Remarks to the Author):

A new experiment has been added based on the previous referee's comment to clarify the mechanism of white reflection, and the referee thinks their paper improved as an interesting one adding the experiments on gender differences and using oil. One main point, the referee thinks the authors need to be careful about the description of oil.

Is the quote from Ref 31 Nixon et al. optimal?

Nature Comm. 11 4108 (2020) Biomimetic design of iridescent insect cuticles with tailored, self-organized cholesteric patterns

Physical Rev. E83, 051917 (2011) Direct determination of the refractive index of natural multilayer systems

The referee thinks those papers are better as a quotation.

However, these refractive index studies are about beetles. Moreover, those are special cuticles with a strong melanin pigment that forms the structural color. Cuticles, which are usually composed of light elements such as carbon, hydrogen, oxygen, may not have a refractive index as high as 1.5. The Oil experiment is a very smart one, but the referee would like the authors to state that you used the beetle value for a high index of refraction and that the actual refractive index of the white area of the olive fruit fly is unknown.

Minor points,

It is unclear what the figure in Fig. S1 means. The referee thinks it would be improved if you could make the figure easier to see and add the explanation of the figure legend.

Line 316 (in Results)

The white spots in males reflects ~16% more visible light than in females. Besides, the variation in reflection by individual females is slightly (1.5 times) higher than by individual males.

Line 371 (in Discussion)

The reflection from white patches in males, which is significantly higher than in females, could be due to the different development of the electron translucent vesicular layer in the two sexes observed over the air sac arborisations, but further investigations are necessary to clarify this point.

Statistically significant?????

Please check small typos such as,

Line 316 reflects – reflect, Line 373 arborisations – arborizations, Line 378 trachee – trachea

REVIEWER #1 (REMARKS TO THE AUTHOR):

- There are still some issues in the introduction. In my review, I gave one example which is considered by the authors ("with periodic features"). However, when I wrote that some sentences in the introduction are easy to be misunderstood, I assumed that the authors would take this polite hint to re-edit the complete text. So, I will be more direct now: Please note that the whole manuscript contains several very general statements which are not scientifically true. An example, line 61 should read "... and requires scattering of all VISIBLE wavelength". One might consider that tiny details, but please do your readers a favor and re-edit the manuscript carefully.

We introduced "visible". We carefully went through the Introduction and did not find any very general statements that do not hold true. We are happy to consider them, if you have any concrete suggestions for improvement.

- The discussion is extended with some more details about potential applications. However, I think the arguments are weak. The industry seems to be very happy with conventional TiO₂ to achieve whiteness. The added patent does not fit to whiteness. It deals with color pigments and is abandoned. You might want to consider that TiO₂ is considered a potential harm for health and the environment recently. That's the reason why many industries look for replacements and other researchers focused on structural whiteness. I have no problem if there is no potential application. The results might be still of great scientific interest. However, if you mention bio-inspiration in the abstract there should be a serious discussion about that in the main text.

We removed any reference to any potential biomimetic applications.

- In my review I wrote: "A smaller issue: I recommend avoiding statements like the one in Line 445 "... for the first time ...". If it's true, the readers will notice by themselves. Your answer is: " If the referee agrees, we would like to keep this statement because we think it helps to highlight some important aspects, such as the novelty of an air sac used in the production of structural color. We would like to stress the functional plasticity of insect organs and not all the readers of a multidisciplinary journal as Biology Communications know the typical function of insect air sacs." I disagree. The term "For the first time" is mentioned in the abstract and on page 20 in the discussion. The phrase "new" is mentioned with the same meaning two times in the abstract, additionally on page 16, and two times on page 20. I assume the readers will get the message if such claims are less frequently found in the manuscript.

We removed some of the redundancies.

REVIEWER #2 (REMARKS TO THE AUTHOR):

A new experiment has been added based on the previous referee's comment to clarify the mechanism of white reflection, and the referee thinks their paper improved as an interesting one adding the experiments on gender differences and using oil. One main point, the referee thinks the authors need to be careful about the description of oil. Is the quote from Ref 31 Nixon et al. optimal?

Nature Comm. 11 4108 (2020) Biomimetic design of iridescent insect cuticles with tailored, self-organized cholesteric patterns

Physical Rev. E83, 051917 (2011) Direct determination of the refractive index of natural multilayer systems

The referee thinks those papers are better as a quotation.

We replaced ref 31 Nixon et al., with the suggested references.

However, these refractive index studies are about beetles. Moreover, those are special cuticles with a strong melanin pigment that forms the structural color. Cuticles, which are usually composed of light elements such as carbon, hydrogen, oxygen, may not have a refractive index as high as 1.5. The Oil experiment is a very smart one, but the referee would like the authors to state that you used the beetle value for a high index of refraction and that the actual refractive index of the white area of the olive fruit fly is unknown.

We added this sentence: “we used the refractive index of the beetle cuticle even if the actual refractive index of the white patches of the olive fruit fly is unknown” in the material and methods section.

Minor points,

It is unclear what the figure in Fig. S1 means. The referee thinks it would be improved if you could make the figure easier to see and add the explanation of the figure legend.

We better explained the Figure as follows

Fig. S1. Reflection spectra smoothing using Savitzky-Golay filter. A typical normalized reflection spectrum from a thoracic white patch of *Bactrocera oleae* male is shown as a grey line. The spectrum was measured at 45° illumination and 0° detection. The spectrum was normalized on the reflection from the white standard (WS1) at 45° illumination and 45° detection. The red line represents the reflection spectrum after smoothing it using Savitzky-Golay filter with 4-th order polynomials over 12 nm windows.

Line 316 (in Results)

The white spots in males reflects ~16% more visible light than in females. Besides, the variation in reflection by individual females is slightly (1.5 times) higher than by individual males.

We replaced reflects with reflect.

Line 371 (in Discussion)

The reflection from white patches in males, which is significantly higher than in females, could be due to the different development of the electron translucent vesicular layer in the two sexes observed over the air sac arborisations, but further investigations are necessary to clarify this point.

Statistically significant?????

Yes, as you can see in the results section:

“The thickness of the vesicular layer of the female is $7.5 \pm 1.9 \mu\text{m}$ ($n= 6$) (mean \pm SD), which is significantly different ($t=4.90$, $d.f.=9$, $p=0.0008$) from that of males that measures $2.9 \pm 0.8 \mu\text{m}$ ($n= 5$) (mean \pm SD).”

In any case, we specified this in the Discussion.

Please check small typos such as,

Line 316 reflects – reflect, Line 373 arborisations – arborizations, Line 378 trachee – trachea

Done